# Assessing Landscape Instability through Land-Cover Change Based on the Hemeroby Index (Lithuanian Example)

**Agnė Jasinavičiūtė \* and Darijus Veteikis**

Faculty of Chemistry and Geosciences, Institute of Geosciences, Vilnius University, 03101 Vilnius, Lithuania; darijus.veteikis@gf.vu.lt
* Correspondence: agne.jasinaviciute@chgf.vu.lt

**Abstract:** The increasing anthropogenic impact on landscapes globally has increased interest in assessing landscape naturalness. This study assessed the changes in land cover in Lithuania and identified the most common reasons for land-use change. Coordination of Environment Information (CORINE) land-cover data were used to assess the changes in land cover naturalness in Lithuania from 1995 to 2018. Land-cover types were ranked according to the hemeroby index, ranging from natural landscape with no human impact to anthropogenic landscapes with excessively strong human impact. Land use trends in Lithuania were related to agricultural and forestry activities. During the analysed period, these areas decreased by as much as 11.19%, and the number of areas with a particularly strong impact from human activities also decreased. Land cover naturalness did increase in areas less suitable for agriculture. The impact of human activities on the naturalness of the landscape needs to be explored in detail at the local level, which should be followed by appropriate spatial-planning decisions to ensure ecological balance through as many sustainable solutions as possible, especially with the European Commission adoption the European Green Deal.

**Keywords:** land-use change; naturalness; hemeroby index; CORINE; landscape assessment

## 1. Introduction

A clean and healthy environment [1,2] and a harmonious landscape that guarantees the right conditions for all living organisms (including humans) to coexist are important resources for all [1,3]. The landscape is heterogeneous, but it is also a dynamic ecosystem, which is constantly changing at different rates depending on the historically and socially determined nature of anthropogenic activities or natural-geographical conditions. Landscape management should therefore be adaptive [4,5]. Indeed, it could be argued that the landscape could be described as a socio-ecological system [3,4]. Many economic, social and environmental problems are related to the rational organisation of territory, so it is especially important to apply as many integrated landscape [6] management methods as possible to bring together different sectors and seek interdisciplinary solutions [7]. The continuous use of natural resources is a strong human impact on the natural environment. Integrated landscape management measures are essential to find ways and means to assess negative impacts [8] in a timely manner and to halt the negative effects of human activities.

Currently, the significance of human activities for the environment is a globally predominant topic [2,9,10]. Excessive use of land, transformation of forests and depletion of biodiversity [11,12] make the landscape less vibrant and more sensitive to economic impacts. The area of eroded and fallow land is increasing and humus in arable soils is decreasing [13]. Anthropogenic impacts [14,15] on the environment are often recorded as negative impacts on the environment, when human economic and social activities lead to environmental degradation, intensive anthropogenic landscape change and biodiversity loss [16].

Landscape conditions have been defined in the literature by many parameters such as sensitivity, degree of damage, capacity, stability and naturalness [17]. Each of these

properties determines the quality of the landscape in some way. To manage and preserve two key interrelated components, i.e., abundance and variety, the indicators of landscape stability and ecological compensation [5,18] need to be thoroughly understood.

Increasing anthropogenic land-cover load has a negative impact on biodiversity [11,19] as well as on landscape resilience. Resilience as a concept applied to ecosystems was first defined as a measure of a system's ability to respond to change and survive disruption [14,20–22]. However, in recent decades, the term has been extended to include not only ecological, but also social, economic, and infrastructural systems [23,24]. Today, the concept of resilience is probably most commonly applied to socio-ecosystems: these are interconnected networks of ecosystems, institutions, actors, and species [25]. Landscape resilience is defined as the ability of a landscape to maintain desired ecological functions in the presence of strong biodiversity and natural landscape processes that are not affected by adverse factors [20].

The concept of landscape naturalness and its assessment have been fairly common objects of research. The naturalness of the landscape is often understood as a natural area that is not affected by human activities. The works that follow this concept [5,26] focus on the preservation of the quality of the natural landscape and biodiversity. However, the terms *hemeroby* or *hemerochora* in areas that have retained their natural character have become more commonly used. The term *hemeroby*, introduced by the botanist Jalas [26,27], is derived from the Greek words *hémeros* (domesticated, cultivated) and *bíos* (life), and this concept was later applied to all ecosystems [26,28]. *Hemeroby* is an integrated indicator used to determine the degree of human intervention in ecological components and ecosystems [28,29]. Based on the hypothesis that human intervention in natural ecosystems causes significant disturbance and, therefore, changes in species composition from the climax to the earlier stages of succession, hemerobiotic status is assessed by estimating the magnitude of this deviation from the climax characterised by potential natural vegetation [18]. Hemeroby, as a concept, is also used for landscape-based analysis, serving as an indicator of the ecological value and landscape diversity, as well as of the extent of the anthropogenic transformation [18,26,27,29]. Hemeroby is commonly used to assess the anthropogenic transformation of phytocenoses and ecosystems [30]. It is, therefore, one of the most commonly used and most convenient assessment criteria also for assessing the naturalness of land cover.

Both the European Landscape Convention and Lithuanian national documents [15] declare that the monitoring of the state of and changes in the landscape is necessary for the rational use of space and its management. Landscape monitoring is carried out in accordance with State Environmental Monitoring programmes. Changes in the structure of the landscape are observed throughout Lithuania, with specific indicators in ecologically sensitive and protected areas (dynamics of the shores of the Baltic Sea, the state of the landscape of the Northern Karst District, and monitoring of state parks). The diversity of the country's landscape, as formed by natural conditions and social activities, was studied and its morphological structure was identified in a study of the diversity of the spatial structure of the landscape and its types in the Republic of Lithuania (2005-2006). Lithuanian Coordination of Environmental Information (CORINE) land-cover data sets for 1995-2018 have been prepared. In 2015, a group of researchers from Vilnius University prepared the paper "Assessment of changes in the structure of the landscape in problem areas at the local level" [31]. However, in the absence of funding, detailed landscape-monitoring work in the country has been discontinued.

Mapping and assessment of ecological situations are becoming very important [3,8,32] for the needs of landscape ecology and territorial planning works. One of the most important tasks in the ecological assessment of the landscape is to assess the reproductive capacity of the landscape components to withstand both physical and chemical anthropogenic stress, and as well as suitability of the landscape mosaic for biodiversity.

This study sought to determine the degree and direction of anthropogenic instability of the landscape in terms of ecological compensation of the landscape and its quality. To

achieve this goal, after evaluating and analysing the international literature and to achieve international compatibility, it was decided to use hemeroby indices following H.P. Blume and H. Sukopp [26,28,29], which reflect the level of human activity in ecosystems. This paper analyses a period of 23 years using CORINE 1995–2018 data. We wanted to assess the main spatio-temporal parameters of the land-cover changes in Lithuania: the frequency, the sequence of land-cover types (direction of change), fluctuations of landscape hemeroby and general landscape instability. A stronger focus on the complex dynamics of land-cover change is essential to better understand the future impacts of man-made land use change on global biodiversity.

## 2. Materials and Methods

### 2.1. Study Area

The object of this research is the territory of the Republic of Lithuania and its landscape (Figure 1). The area of the country is 65,286 km$^2$. The counties containing the largest cities—Vilnius, Kaunas and Klaipėda—are the most populous in Lithuania. At the beginning of year 2020, 2 million people lived in Lithuania, with a population density of 42.8 people per square kilometre [33]. According to the Land Information System, operated by the National Agricultural Service under the Ministry of Agriculture, the distribution of the land fund is various (Figure 1). The Lithuanian landscape is dominated by forests (33%) and water bodies (46%); other land uses are evenly distributed [34]. Most of Lithuania's forests are coniferous, with pine forests accounting for 34.6% and spruce forests accounting for 20.9% of the total area. Of the deciduous forests, birch (22.2%) and black alder forests (7.6%) are the most abundant. Almost a third of all Lithuanian forests are located within the boundaries of protected areas or their buffer zones.

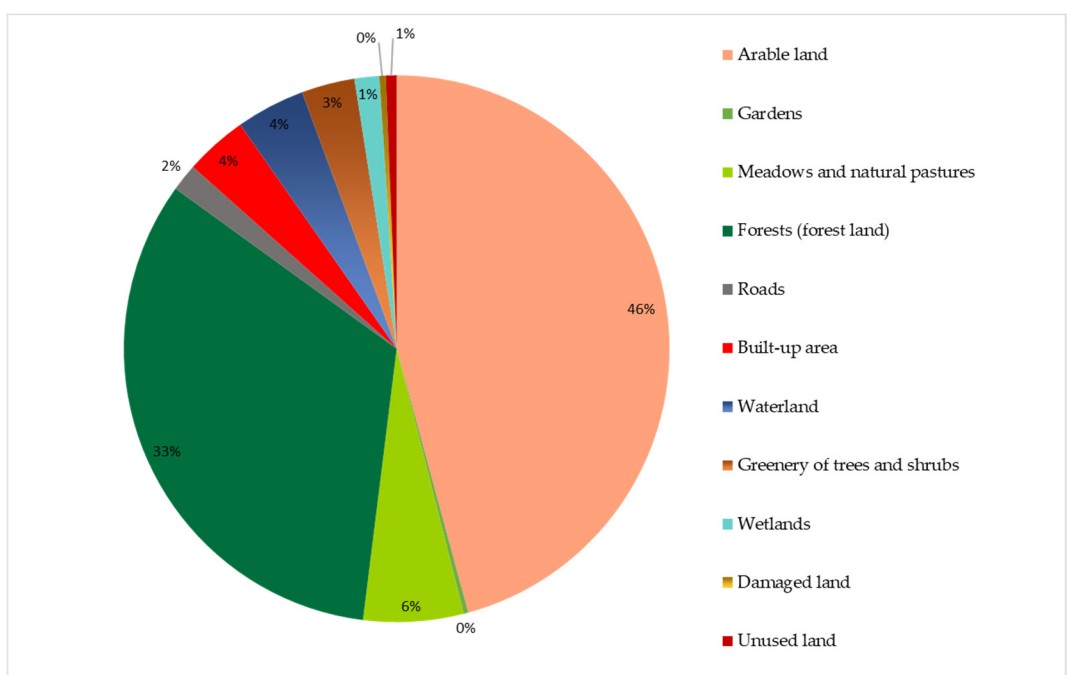

**Figure 1.** Study area and distribution by land use in Lithuania (Data from Land Information System https://zis.lt/statistika/zf/, (accessed on 2 April 2022).

The average annual rainfall in Lithuania for the period 1961–2018 is around 695 mm, but there has been a significant increase in annual rainfall of 114 mm, or 17 per cent. The rainiest year in Lithuania was 2017, when the average annual rainfall in Lithuania was 875 mm. The biggest increase in precipitation is in winter, with a decrease in spring. On average, annual soil temperatures rose by 2.6 °C between 1961 and 2018. As the climate

changes, deeper soil temperatures are also rising, with annual soil temperatures at depths of 0.2 m, 0.8 m, 1.6 m and 3.2 m increasing by 1.6 °C between 1961 and 2018.

Climate change is changing the dates of the first and last snowfall. The date of the first snowfall is found to have shifted by 10 days on average between 1961 and 2018. The last snowfall in spring occurs on average 19 days earlier. As the climate changes, not only is the thickness of snow cover decreasing, but also the likelihood of winter precipitation.

The recurrence of natural disasters in Lithuania was found to have increased significantly. The number of such events has increased by an average of 5 over the last 38 years (1981–2018). The largest increases were in the number of very high winds, very heavy rain and very severe storms.

All the processes of relief formation and transformation are manifested differently in different parts of Lithuania. The landscape of clayey plains and undulating plateaus, as well as moraine hills cover more than half of Lithuania. The landscape of sandy plains, valleys, moraine ridges and lakes are also quite common, but unequally distributed throughout the country. The rarest are the unique landscape types of spit (represented by the Curonian Spit); erosive horns and ravines, and; deltas, coastal plains and lagoons [35,36]. By cultivation type, generally, the agrarian landscape prevails in Lithuania, which is related to the high prevalence of clayey, rather fertile plains (Figure 2). The most preserved landscape, characterised by forest cover, occupies an intermediate position in the country's territory, and the urbanised landscape type takes, relatively, the smallest share.

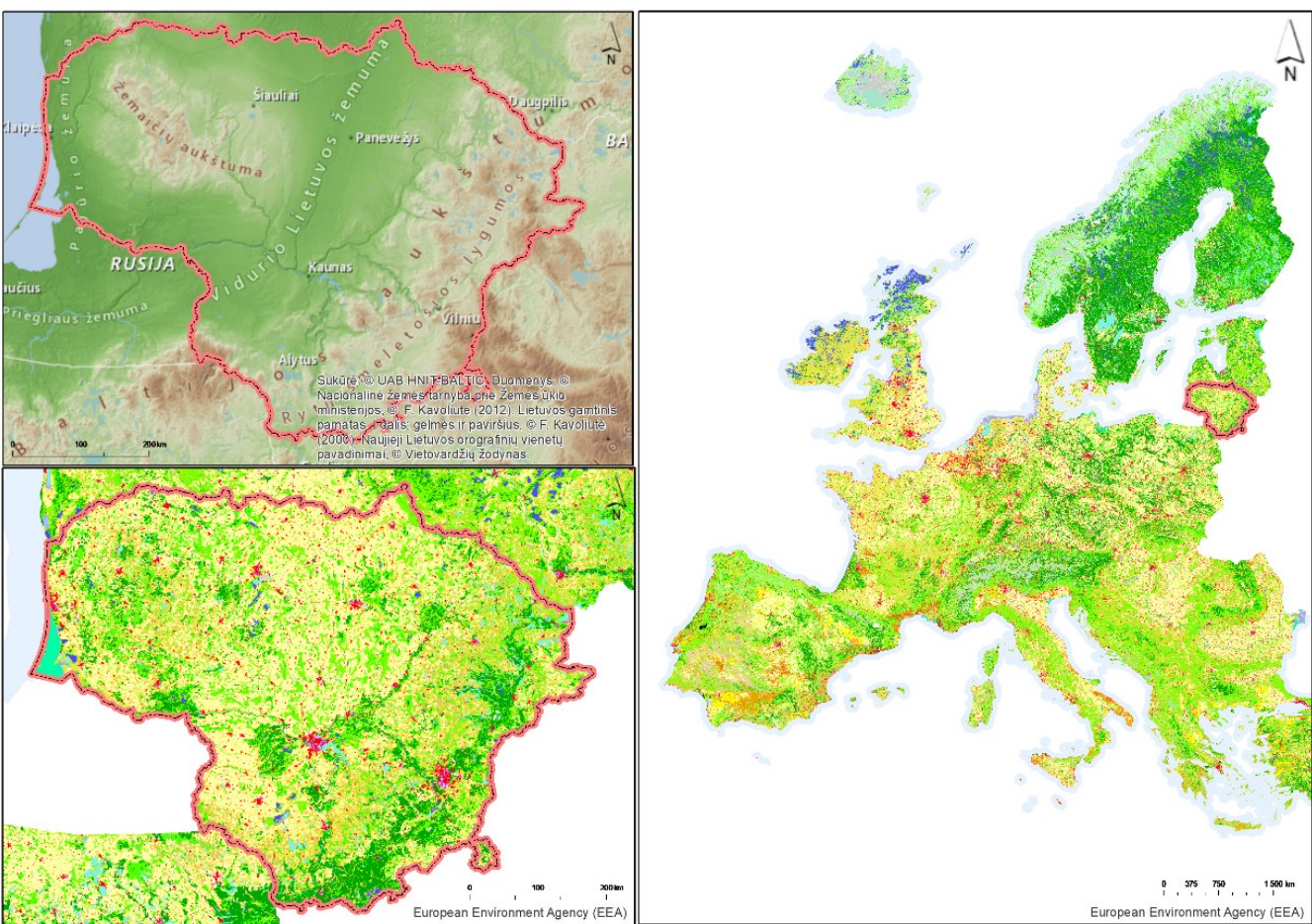

**Figure 2.** Study area. Physical map of Lithuania showing elevation (**top left**) and distribution by land use (**left**) and land use in Europe according EEA CORINE Land Cover 2018 data (**right**).

Lithuania has a National Landscape Management Plan (NLMP), a state-level spatial planning document covering the country's entire territory (except for the country's terri-

torial waters in the Baltic Sea), which sets out the principles for the protection and use of the country's landscape and the most important management directions. Few countries in the world have such landscape management plans. Lithuania, which ratified the European Landscape Convention in 2002 (which establishes the general rules of landscape design and helps in setting reasonable goals and measures for its quality), is at the forefront of landscape planning and is striving to ensure that its valuable landscapes are responsibly managed, used and protected.

## 2.2. Data and Territorial Units for Analysis

Data on the Lithuanian land cover (i.e., the CORINE land-cover data) were used as the basis to assess the anthropogenic instability of the land cover. The level of naturalness (*hemerobia*) of the landscape was determined using the CORINE land-cover data sets from 1995, 2000, 2006, 2012 and 2018 prepared by the Copernicus Land Monitoring Service. CORINE land-cover data provide an opportunity to monitor the development trends of various processes, such as: urbanisation and its infrastructure development, agricultural areas, forests and other natural areas, wetlands and water bodies in Lithuania. CORINE land-cover types are coded using the standard CORINE classification, which distinguishes 3 levels (L1, L2, L3). The L3 level, which consists of 31 land cover classes, was used in the calculations. The smallest mapped object in these databases is 25 ha for area objects and 100 m for linear objects. The smallest mapped object in the database is 5 ha for area objects, while maintaining the minimum object width of 100 m.

The initial territorial units for the calculations of the indices were those obtained from the intersection of the five CORINE data sets from 1995, 2000, 2006, 2012 and 2018. These were the elementary units given hemeroby values for each time section. The obtained units were quite diverse in size (0.01 ha to 891.87 ha) and shape.

Based on their useful features for geospatial analyses [37] and following similar methodologies and applications previously used [38,39], a network of 15,307 hexagons was created to cover the territory of Lithuania to ensure even and generalised calculations of land-cover changes, thus yielding a hexagonal grid for statistical analysis to evaluate homogeneous spatial changes [39,40]. The area for one hexagon was set to 4.39 km$^2$ (the length of one side of a hexagon 1.3 km). The selected area guaranteed a sample of representative landscape land cover areas in one hexagon and an even coverage throughout the analysed area.

## 2.3. Methods of Assessing the Landscape Instability

To determine the naturalness of landscape areas and the magnitude of the human-induced deviation from possible naturalness in a particular ecozone, it is first necessary to provide index (or coefficient) values for different types of land cover that can be used in quantitative analysis. Such attempts have been made, but expert indices are often subjective and debatable. This is illustrated by the fact that, in Lithuania alone [35,36,41–43], different naturalness indices by several authors are given for the same land-cover types.

After the literature analysis and for the sake of international compatibility, it was decided to use a hemeroby index according to the work of H.P. Blume and H. Sukopp [28], which reflects the level of human activity in ecosystems (Table 1).

The selected range of the hemeroby index was that most commonly used in the literature [28,44], thus ensuring comparability with other studies of this kind. The intensity, duration, and range of human exposure were considered in assigning a hemeroby value to the land cover classes [39,45–47].

**Table 1.** Naturalness and hemeroby indices of land-cover types in Lithuania according CORINE data. The degree of naturalness (hemeroby index) adopted from [19,28,44].

| | Degree of Hemeroby Description | Level 3 of Land Cover Classification (L3): 100,000 |
|---|---|---|
| Hi1 | Ahemerobe. Almost unaffected by man, natural | *Not indicated in Lithuania* |
| Hi2 | Oligohemerobe. Weak impact of human activities, close to nature | 311 Broad-leaved forest<br>312 Coniferous forest<br>313 Mixed forest<br>331 Beaches, dunes, sands<br>411 Inland marshes<br>412 Peatbogs<br>521 Coastal lagoons<br>522 Estuaries<br>523 Sea and ocean |
| Hi3 | Mesohemerobe. Moderate exposure to human activity | 321 Natural grassland<br>322 Moors and heathland<br>324 Transitional woodland/shrub<br>333 Sparsely vegetated areas<br>334 Burnt areas |
| Hi4 | β-euhemerobe. Moderately strong effects of human activities, far from nature | 141 Green urban areas<br>231 Pastures<br>243 Land principally occupied by agriculture, with significant areas of natural vegetation<br>511 Water courses<br>512 Water bodies |
| Hi5 | α-euhemerobe. Strong effects of human activities | 142 Sport and leisure facilities<br>211 Non-irrigated arable land<br>242 Complex cultivation patterns |
| Hi6 | Polyhemerobe. Very strong impact of human activities | 112 Discontinuous urban fabric<br>131 Mineral extraction sites<br>132 Dump sites<br>133 Construction sites |
| Hi7 | Metahemerobe. particularly strong effects of human activity | 111 Continuous urban fabric<br>121 Industrial or commercial units<br>122 Road and rail networks and associated land<br>123 Port areas<br>124 Airports |

Thus, using the hemeroby index with scientific recognition, before performing spatial data calculations for Lithuanian CORINE land-cover types in 1995–2018, corresponding indices of land cover naturalness were provided for the data sets for each period. The hemeroby data were mapped using ArcGIS 10.2.1 software. The following steps towards the overall assessment of landscape instability were performed.

*(1) Sequence (direction) of land-cover change*. A linear calculation of the direction factor was used to calculate the direction of land-cover change [24]. Depending on the level of the land cover classifier, an appropriate hemeroby index was obtained. The calculation of the direction of land-cover change was performed using Formula (1).

$$\Delta H = (CLC00-CLC95) + (CLC06-CLC00) + (CLC12-CLC06) + (CLC18-CLC12) \quad (1)$$

where $\Delta H$ is the direction coefficient for land-cover change, which can be positive (+; i.e. the changes that have taken place are strongly influenced by human activity), or negative (-; i.e., the changes that have taken place are natural, weakly or not at all affected by human activity); CLC95 is the Land Cover Naturalness Index 1995; CLC00 is the land cover

naturalness index in 2000; CLC06 is the land cover naturalness index in 2006; CLC12 is the land cover naturalness index in 2012; CLC18 is the land cover naturalness index in 2018.

The coefficient of change of the direction of change in each landscape hexagon was obtained. If the number is positive, then the change has been to the natural side; if the number is negative, then the change is to the anthropogenic side. In the following calculations, only the direction sign (minus or plus), which shows the direction of change, was used.

*(2) Fluctuations in land-cover change.* Naturally, the geo-ecological balance and natural connections of the landscape are disturbed by changes in the landscape. However, capturing the range of landscape hemerobiotic variations has a certain importance for understanding the depth of human impact, as well as natural abilities for landscape restoration and their interchange within the research area. The idea was to detect the amplitude of the hemerobiotic fluctuations in the landscape. Amplitude is a measure of the variation of a fluctuating magnitude in physics, showing the change in the value of a magnitude during fluctuations.

The following formula was used to determine the amplitude of landscape (hemeroby) change fluctuations (2). The formula was designed to show the averaged deviation of hemeroby from the mean value to both sides of the mean, thus displaying the averaged fluctuation of hemeroby in a spatial unit during the five stages of the CORINE data collection.

$$\text{Ksa} = \left( \sqrt{ \left( \frac{(\text{CLC95} - \sqrt[2]{\text{average}}) + (\text{CLC00} - \sqrt[2]{\text{average}}) + (\text{CLC06} - \sqrt[2]{\text{average}}) + (\text{CLC12} - \sqrt[2]{\text{average}}) + (\text{CLC18} - \sqrt[2]{\text{average}})}{5} \right) } \right) * 2 \qquad (2)$$

where Ksa is the amplitude coefficient of land-cover change fluctuations; CLC95–CLC18 are the land cover naturalness indeces in 1995–2018 (as in Formula (1)); average is the average value of the landscape naturalness indices is calculated by adding the naturalness values from the years 1995–2018 and averaging them.

*(3) Frequency of land-cover change.* In man-made ecosystems, land-cover changes can occur at different speeds and frequencies due to ever-changing environments, as well as to socio-economic conditions [18,35]. The frequency of change illustrates the number of times the land cover has changed over a period of time, while also indicating the speed at which these processes occurring. The concept of the frequency of land-cover change can also be based on an analogous frequency of disturbance underpinning the study of the dynamics of ecological communities and is incorporated into various theoretical frameworks [3,48,49].

The frequency of land-cover changes in a spatial unit is the share of the periods that are marked by changes in the hemeroby within the total number of the periods studied (Formula (3)).

$$F = N_{ch}/N \qquad (3)$$

where F is the frequency of land-cover change; Nch is the number of the periods with changes in hemeroby; N is the total number of periods studied.

*(4) Geospatial recalculation of sequence (directions)*, fluctuations and frequency of land-cover change index values within the hexagon network. The total amount of change was multiplied by the land-cover area. The values were then combined using ArcMap's summarise function to estimate the overall rate of land-cover change in the hexagon, and the resulting value was divided by the area of the hexagon (i.e., 4.39 km$^2$).

*(5) Landscape instability.* After detecting the direction of land-cover change, the amplitude of the fluctuations and the frequency of change within the hexagons, classification was performed according to the principle that the instability of the landscape in both directions (nature restoration or anthropogenisation) grows with the increase in fluctuation magnitude and the frequency of changes. After obtaining the indicators of all parameters, a matrix of values of the landscape instability indicator was formed (Table 2).

**Table 2.** Landscape anthropogenic instability matrix.

| | | Frequency of Change with a Sign of Direction of Change | | | | | | | | |
|---|---|---|---|---|---|---|---|---|---|---|
| | | >1.01 | 0.51–1.00 | 0.26–0.50 | 0.01–0.25 | 0 | 0.01–0.25 | 0.26–0.50 | 0.51–1.00 | >1.01 |
| | | to Anthropogenization "+" | | | | (no Visible Changes) | | to Naturalness "−" | | |
| Landscape fluctuations | 0 | 0 | 0 | 0 | 0 | 0 | 0 | 0 | 0 | 0 |
| | 0.01–0.50 | 2 | 1 | 1 | 1 | 0 | −1 | −1 | −1 | −2 |
| | 0.51–1.00 | 2 | 2 | 1 | 1 | 0 | −1 | −1 | −2 | −2 |
| | 1.01–1.50 | 3 | 2 | 2 | 1 | 0 | −1 | −2 | −2 | −3 |
| | >1.51 | 3 | 3 | 2 | 2 | 0 | −2 | −2 | −3 | −3 |

All identified changes in naturalness from intensive land cover renaturalisation to intensive anthropogenisation, depending on the frequency and sequence of change, as well as the fluctuations of change, were divided into seven instability classes and concisely described (Table 3). The result of land-cover changes was assessed: the more frequent the changes in the area and the greater the fluctuations of the changes, the less stable the landscape and the more likely that many processes take place in it. The end result can be twofold: renaturalisation, where the biomass of the forest and shrubs increases, the growth of forest and shrubs occurs in the areas of agricultural areas, natural succession takes place at the sites of former felling, and the processes of swamping are recorded; or spatial development (urbanisation), that is, the transformation of natural or agricultural areas into urbanised areas (of varying density of construction, industrial, commercial or infrastructure areas).

**Table 3.** Landscape instability classes and their description.

| Landscape Instability Class | Description of the Landscape Instability Class |
|---|---|
| −3 | intense change with large fluctuation amplitude, rapid change (3 or more changes), strong renaturalisation. |
| −2 | mean change with mean amplitude of fluctuations, change in mean velocity (2 changes were observed during the analyzed period), renaturalisation. |
| −1 | slow change with a small amplitude of fluctuations, only one change in land cover occurred during the analyzed period. |
| 0 | changes unrecorded or very slight change ranging from minor changes in renaturalisation or anthropogenisation. A mixed change can be recorded. |
| 1 | slow change with a small amplitude of fluctuations, only one change in land cover occurred during the analyzed period. |
| 2 | average change with average amplitude of fluctuations, change of average speed (2 changes were observed during the analyzed period). |
| 3 | intense change with large amplitude, rapid change (3 or more changes), strong anthropogenisation. |

During the period under review, Lithuania experienced many land-cover changes of varying magnitude. However, the most frequent and significant changes were:

- Agrarian change—conversion of natural areas into agricultural areas' transformations that have taken place within a class of agricultural areas;
- Deforestation (forest change, including clear-cutting)—transformation of forest areas into transitional forests and shrubs due to felling and forest-biomass loss;
- Renaturalisation—the increase of forest and shrub biomass, manifested by forest and shrub growth; on the site of agricultural areas, former felling sites (natural succession), swamping processes, and;
- Urbanisation—transformation of natural or agricultural areas into urbanised areas (areas of various densities of industrial and commercial, infrastructure, etc.).

## 3. Results

### 3.1. Sequence of Land-Cover Changes, 1995–2018

Assessing the data for the 23-year period, the structure of the Lithuanian landscape appears to be is changing (Table 4). There are areas where natural structures and, naturalness prevail throughout the period under analysis, and the changes that take place are very small or non-existent. The main processes recorded in the analysed area are related to spontaneous afforestation.

**Table 4.** Land cover naturalness changes in Lithuania, 1995–2018. Hi1–Hi7 indicates the degree of naturalness (hemeroby index; according to Table 1) according to CORINE data. There are no recorded areas with hemeroby index Hi1 (i.e., no completely natural sites). An upward green arrow indicates an increase in land cover between 1995 and 2018, while a downward red arrow indicates a decrease in land cover.

| Hemeroby index | 1995 (km²) | 2000 (km²) | 2006 (km²) | 2012 (km²) | 2018 (km²) | Change (km²) 1995–2018 | Change (%) 1995–2018 | |
|---|---|---|---|---|---|---|---|---|
| Hi1 | 0 | 0 | 0 | 0 | 0 | 0 | 0 | |
| Hi2 | 22,298.62 | 20,262.91 | 20,222.73 | 23,042.49 | 23,125.34 | 826.72 | 3.70 | ⬆ |
| Hi3 | 1845.09 | 2331.42 | 2402.15 | 3397.15 | 3287.41 | 1442.32 | 78.17 | ⬆ |
| Hi4 | 12,220.07 | 11,012.98 | 10,947.19 | 11,175.19 | 10,852.76 | −1367.31 | −11.19 | ⬇ |
| Hi5 | 30,908.96 | 30,911.96 | 30,925.52 | 29,182.76 | 29,512.72 | −1396.24 | −4.52 | ⬇ |
| Hi6 | 1596.85 | 1565.08 | 1581.51 | 1627.40 | 1642.96 | 46.11 | 2.89 | ⬆ |
| Hi7 | 480.97 | 470.31 | 475.55 | 429.15 | 432.95 | −48.02 | −9.98 | ⬇ |

Areas such as green urban areas, pastures, arable land with natural vegetation inclusions and the shores of water bodies) underwent major changes. During the whole analysed period, the areas of moderate human exposure (Hi4) decreased by as much as 11.19% (see Table 4 and Figure 3). There has, however, also been a decline in areas of particularly high human impact (Hi7). Such a change can be attributed to the rather intensively implemented landscaping projects [50,51] in Lithuania, during which damaged areas were reclaimed, for example, former quarries were recultivated and unused buildings were demolished to increase the aesthetic value of the landscape.

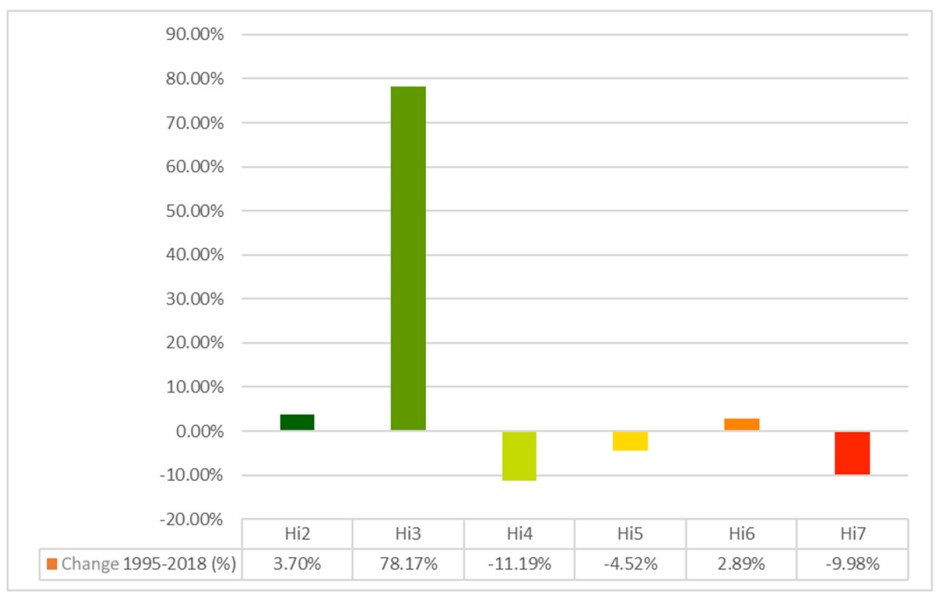

**Figure 3.** Overall land cover naturalness change in Lithuania, 1995–2018 (differentiated by percentage change).

Assessment of the change in the naturalness of the Lithuanian landscape from 1995 to 2018 showed the strongest growth in the areas with a hemeroby index value of Hi3—that is, an area with moderate human exposure. This naturalness index is given to areas such as natural meadows, wastelands and heaths, as well as transitional stages of the forest, which can be related to the process of renaturalisation. Hi2 also increased during the analysis period. According to land-cover data, the area affected by intensive and technogenic human activities has increased by just over 800 km². This naturalness index value is attributed to forest areas, as well as to peatlands and water bodies, so changes in these lands can be related to the plantation of new forests or spontaneous overgrowth, as well as the restoration of hydrological status [52] in damaged wetlands in recent years.

The results obtained after the calculations of the direction (sequence) of land-cover changes (Figure 4) illustrate that, in most of the territory of Lithuania (just over 70%), the changes were insignificant (ΔH from 0 to 0.76). Changes in land cover related to human economic activities followed by the loss of natural features took place in 19.72% of the country's territory (ΔH>0.77). Changes in anthropogenisation were mainly concentrated around major cities, as well as in the central and northern parts of the country.

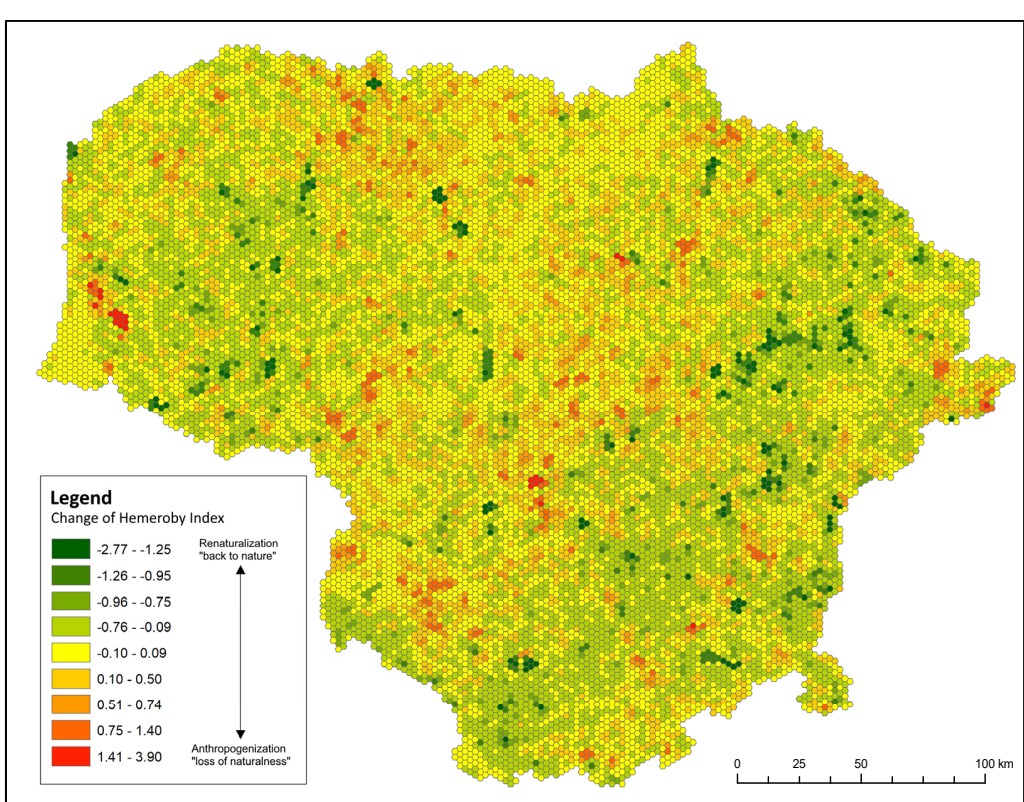

**Figure 4.** Land-cover change trends (tendencies) 1995–2018.

Changes in land cover towards naturalness (ΔH<−1.26) occurred in 4.29% of the area and were mainly in areas with more difficult terrain, such as the highlands of Samogitia (western Lithuania) and Aukštaitija (eastern Lithuania). Although processes influenced by anthropogenic activities took place in these areas, areas where renaturalisation took place rapidly are emerging.

### 3.2. Fluctuations of Land-Cover Change in the Period 1995–2018

Taking into account the amplitude of changes in the naturalness index of the land cover, the following categories of fluctuations in the hemeroby index values can be distinguished: insignificant fluctuations (ΔH between 0 and 0.57), medium size fluctuations (ΔH between 0.58 and 0.76) and significant fluctuations (ΔH between 0.77 and 3.90). About

40% of the analysed areas belong to the first category. Only 5% of the country's territory underwent very sudden or frequent changes in land cover. Such areas have completely lost their naturalness.

The situation of land-cover change fluctuations is presented on the map in Figure 5, which highlights the areas where the most pronounced hemerobiotic changes, with the strongest tendencies to repeat, have taken place.

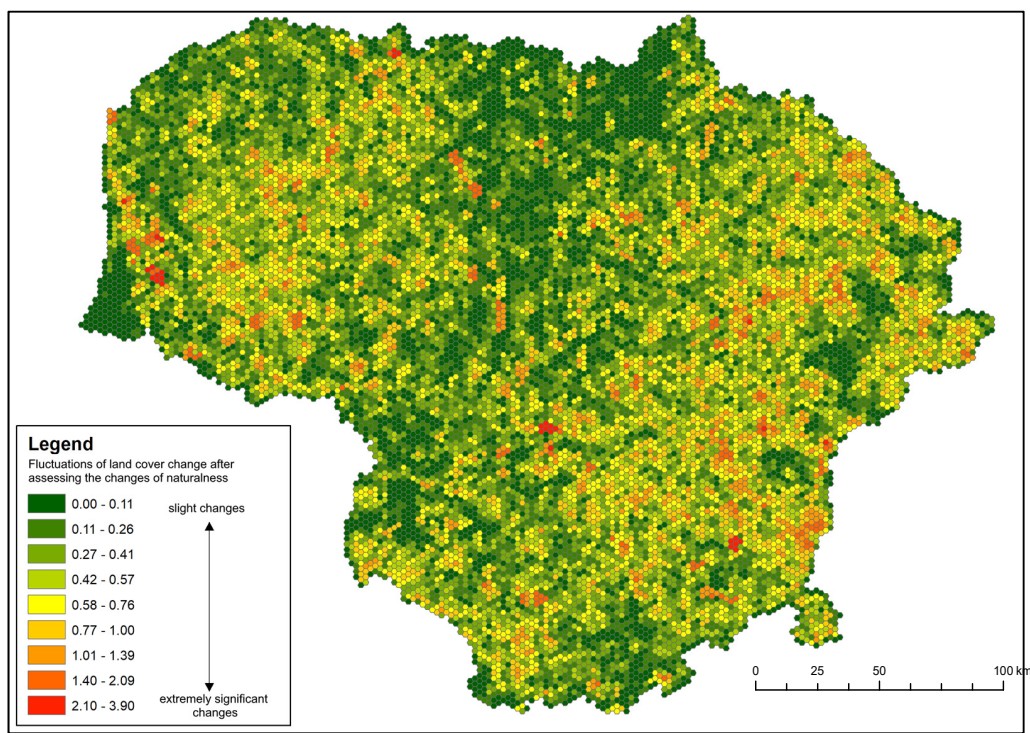

**Figure 5.** Fluctuations in land-cover change in Lithuania, 1995–2018.

### 3.3. Frequency of Land Cover, 1995–2018

The frequency of land-use change refers to the number of times the land use has changed over a period of time. The CORINE land-cover data analyzed in this work make it possible to see whether changes took place over a period of five years. This indicator shows the transformation of land use between land use types and expresses its intensity in the territory, which makes it possible to understand the spatial heterogeneity of land use processes. A maximum of four changes could have taken place during the analysed period 1995–2018; however, a relatively large part of the areas showed no recorded land-cover changes (Figure 6).

No change during the whole analysed period (1995–2018) was recorded in 15,283.45 km² of the territory of Lithuania, that is, in about 23.41% of the total area of the country. This change in stability can be attributed to some of the stricter regulations in the area where no economic activity is possible and natural processes are less pronounced for example, strict protected areas.

In some parts of analysed areas, the frequency is not rapid even though the area of change is large and could substantially affect land use.

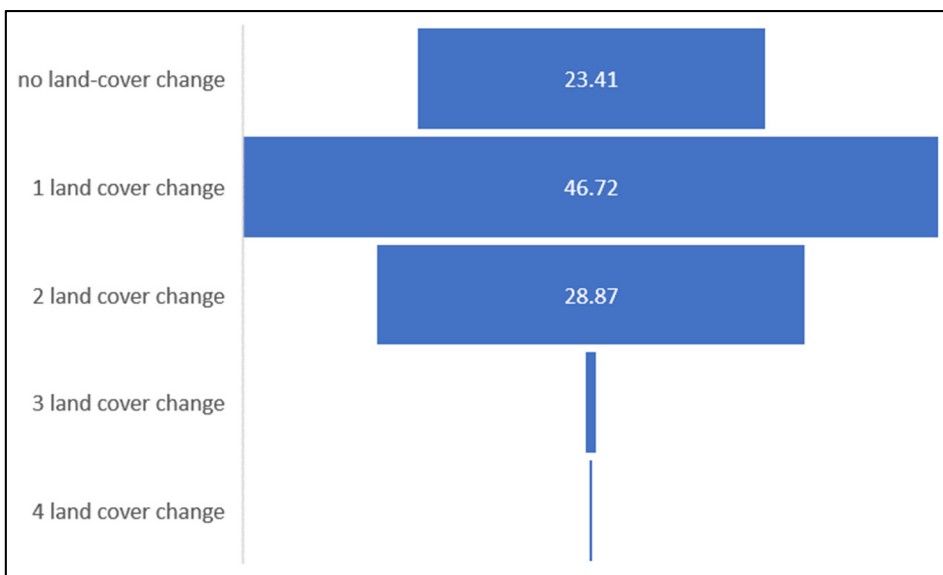

**Figure 6.** Percentage change of land-cover change-frequency distribution between 1995 and 2018.

*3.4. Landscape Instability in Lithuania, 1995–2018*

In this work, the assessment and mapping of landscape stability was performed according to the complex evaluation of land-cover change direction, amplitude and change frequency, all of which were integrated into a matrix of landscape instability. Later, based on the distribution of landscape instability classes, the landscape types provided in the NLMP were described.

Considering the naturalness and temporal features of landscape changes, seven landscape instability classes were identified, which were unevenly distributed in the analysed area (Table 5 and Figure 6) over the 23 years of the study period. Areas where land-cover change has not taken place and the landscape has remained stable (class 0) accounted for only 2.25% of the country's territory, including the Curonian Lagoon, where no structural changes have been recorded. The remaining part of the analysed territory followed two paths of change: nature oriented (classes −1, −2 and −3) and anthropogenic (classes 1, 2, and 3).

**Table 5.** Landscape instability in Lithuania, 1995–2018.

| Landscape Instability Class | Area km$^2$ | Part of the Territory of the Country % |
|:---:|:---:|:---:|
| −3 | 526.8 | 0.78 |
| −2 | 10,268.21 | 15.28 |
| −1 | 28,153.07 | 41.90 |
| 0 | 1510.16 | 2.25 |
| 1 | 20,720.8 | 30.84 |
| 2 | 5342.63 | 7.95 |
| 3 | 676.06 | 1.01 |
| Total: | 67,197.73 * | 100 |

* The area of the Republic of Lithuania is the total area of the territory according to the 1: 200,000 georeferenced databases, on the basis of which a hexagonal network was prepared.

Areas with an instability class −3 were unevenly distributed in Lithuania and accounted for only 0.78% of the country's total area. Renaturalisation processes prevail in these areas, along with agrarian change, when agricultural land is no longer used for its intended purpose; such areas are later accounted as abandoned.

Areas in class −3 are mainly located in wetlands and urban landscapes (Figure 7). Areas with medium instability (class −2) and more than two changes in the land cover during the analysed period occupied 15.28% of the country's territory. The most common

conversions in these areas are agrarian change and renaturalisation. Deforestation and urbanisation processes have been recorded in smaller areas within the country, but were almost evenly distributed across the country.

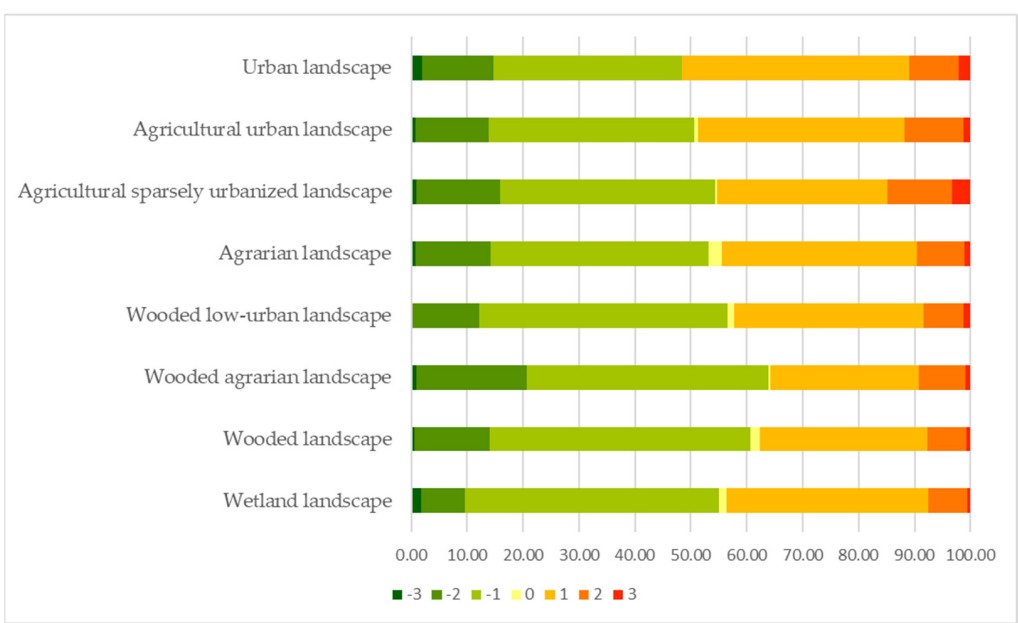

**Figure 7.** Distribution of landscape instability classes in areas of different landscape types (according to the National Landscape Management Plan).

Most of the instability areas in the −2 class were located in areas of forested agrarian (19.72%) and agrarian sparsely urbanised (14.95%) landscape types. During the analysed period, the largest spatial extent belonged to the class (−1) in which changes occurred slowly moving towards restoration of natural qualities of landscape, with a small amplitude and very low frequency of change. Over a period of 23 years, there have been minor changes in such areas. Most of these changes have taken place in wooded (46.70%), wetland (45.49%), wooded agrarian (43.27%) or wooded sparsely urbanised (44.53%) landscape areas. A large portion of such relatively stable landscape has also been recorded in areas with agrarian (39.01%), agrarian sparsely urbanised (38.57%), agrarian urbanised (36.76%) or urbanised (33.59%) landscapes.

The anthropogenic instability was mostly slow (class 1), with a faint amplitude of fluctuations (30.84% of the country area), and this pattern of changes was dominant. These areas were almost evenly distributed throughout Lithuania (Figure 8), except for the highlands of Samogitia, the Baltics and Sūduva. In these areas, the most pronounced processes of landscape change were associated with the process of deforestation, as well as the conversion of natural areas to agricultural land and the process of urbanisation. Class 1 instability areas predominated in urban (40.65%) and urbanised (37.00%) land covers. Class 2 instability was characteristic of 7.95% of the territory of Lithuania. This class was most pronounced in areas with agrarian sparsely urbanised (11.60%) and agrarian urbanised (10.48%) land cover. Meanwhile, changes in Class 3 instability occurred at 1.01% more than three times.

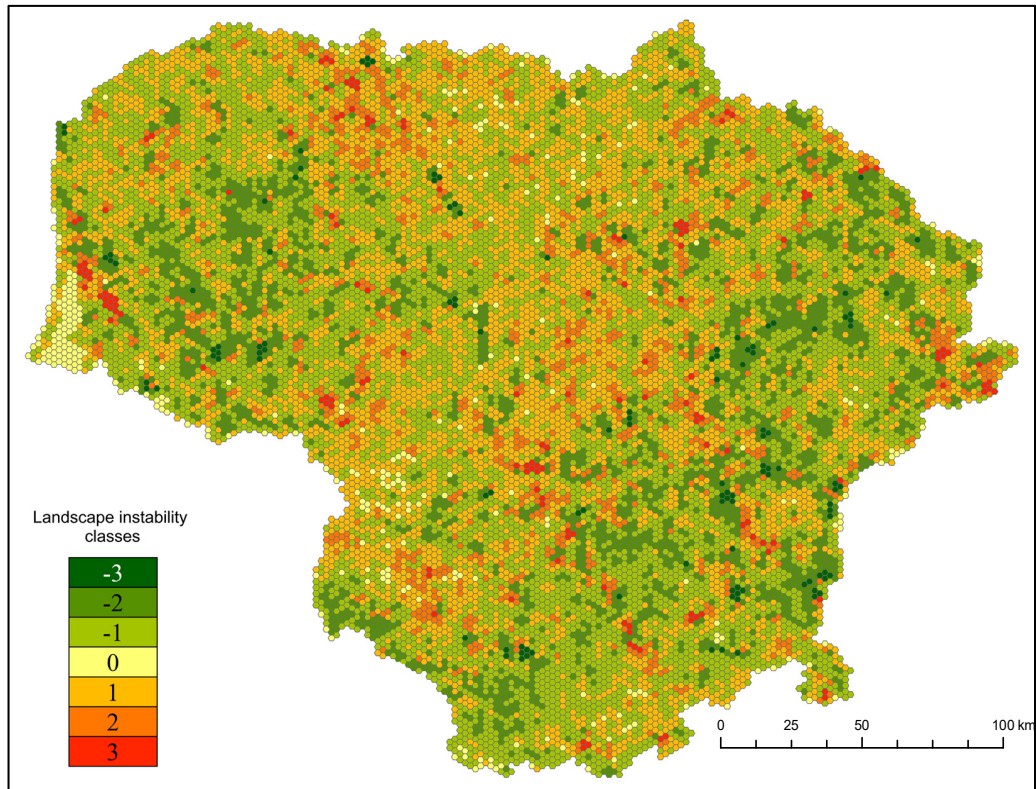

**Figure 8.** Landscape instability in Lithuania, 1995–2018. Landscape instability classes represented in Table 3.

## 4. Discussion

Landscape stability is one of the most important parameters determining the state of the environment and the changes that take place in it under the influence of natural and anthropogenic factors [4,10,14,26,31,47]. The nature of land-cover change depends on the location in the geographical environment and the characteristics, nature and extent of the anthropogenic impact. Landscape stability is the ability of a geosystem to maintain its structure and mode of operation under changing environmental conditions.

Assessment of land-cover change, although often the only viable alternative in the absence of qualitative and consistent data on land-cover change, do not allow a direct correlation with changes in a particular area in this study. A similar conclusion is presented by other authors [53] who have studied the land change in the Lithuanian landscape and tried to identify the main reasons for it.

One of the most important challenges is reconciling urbanisation and nature conservation so as not to reduce significant areas of the country's natural landscape. The impact of human activities on the naturalness of the landscape needs to be explored in detail at various levels and should be followed by appropriate spatial-planning decisions to ensure ecological balance through as many sustainable solutions as possible, especially when the European Commission adopts the European Green Deal [51] as a set of proposals to transform the EU into a modern, resource-efficient and competitive economy.

Before finalising the outcome, some functional methodological obstacles inherent in the techniques used in the innovative approach must be addressed. As far as we know, there is no popular method for comparing sub-focal phenomena, because the spatial device used for scoring may be different along with the rating scale. It should be noted that the assessment of landscape characteristics, in line with the degree of human influence, varies among researchers [4,8,29,31,35,41,46,53], leaving the general idea of landscape for a more focused dialogue. Assigning a specific land-cover type to a separate naturalness index value can also be subjective. Studies involving landscape change assessment have

been carried out in Lithuania [26,31,35,41,43,53], but they have not used the hemeroby index. This is not a definitive statement about the naturalness of the Lithuanian landscape. The main purpose of this study was to assess landscape instability and show the impetus behind those changes. We decided to use a standardised input data set and index values for the land-cover type. The Hemeroby index is a very generalised index that does not just summarise the results of area statistics into one value [54,55]. It also includes an assessment of the extent of human transformation of the landscape, as well as the human impact on vegetation and the geosphere as a whole.

Since 1995, there has been a minimal and statistically negligible improvement in the natural land cover nationwide (Figure 4). In locations with better agricultural conditions, the naturalness index has dramatically reduced. Conversely, the landscape naturalness index has increased in regions with a significant concentration of forested areas unsuited for cultivation (Figure 8). Due to the peculiarities of agriculture and forestry, current land-cover structures inherently affect the landscape's naturalness.

The proposed approach to assess landscape anthropogenic disturbance in Lithuania and its spatial distribution was based on the integration of temporal characteristics of hemeroby index fluctuation (direction, fluctuation range and frequency) within the chosen landscape units. Our results illustrate that changes can be caused by both the natural processes of renaturalisation and anthropogenic processes (Figure 7). In those areas where there has been no land-cover change, the landscape ecosystems are sufficiently stable, although various seasonal variations are known, but their impact is insignificant. This can also be related to the protected areas [15] in those locales (i.e., state parks and state nature reserves), which promote the restoration of natural ecosystems and ensure the balance of the landscape.

Lithuanian spatial-planning documents enshrine the concept of preservation and protection of the natural landscape [54] in the shape of a pan-national spatial structure (i.e., the Nature Frame), but the effectiveness of the operation of this system is not monitored. This is very important because the Nature Frame connects all nature reserves with other ecologically valuable or relatively natural areas, on which the overall stability of the landscape is based to form a landscape system of geo-ecological compensation zones. The results (direction, fluctuations and frequency of land-cover change) of this work could be used to support the Nature Frame and measures for improving its effectiveness. These results can be used to make plans for the problem areas in the Nature Frame, by detailing the management of these areas and by taking real action, such as promoting organic farming, crop rotation, introducing ecologically friendly measures, implementing good agricultural practices, expanding green infrastructure, and establishing additional protected areas [13,15,41,54]. Maintaining the structural integrity of the Nature Frame requires practical action to preserve environmental stability, historical and cultural values, maintenance of damaged areas and a focus on quality. In the context of the legal framework, it is necessary to continuously improve the legislation relating to the Nature Frame and to develop the necessary supporting methodologies.

Figure 6 illustrates the distribution of instability in Lithuanian land cover during the period 1995–2018. Fluctuations in naturalness, when the ongoing changes are related to areas with a high natural index, are located in the highlands—the Baltic Highlands, the Samogitian Highlands, and the Sūduva Highlands. The highlands are not conducive to agriculture, so natural land-use change is predominant, but there are also cases of declining forest biomass during deforestation.

## 5. Conclusions

The landscape was quite stable in 23.41% of the investigated area; in the rest of the country, the landscape was unstable and underwent changes mainly due to the emergence of natural areas through renaturalisation, agrarian land abandonment and spontaneous restoration of natural (woody) vegetation such as forests, young forests and shrubby meadows. The quantitative evaluation of land-use change revealed differences over the study

period. From 1995 to 2000 the land-use change trend was towards anthropogenization, while moving towards 2018, it took a big step back towards naturalisation. These facts indicate the quite intensive land-cover fluctuations that are also characteristic of the rest of Lithuania.

In general, the land cover naturalness in Lithuania has tended to improve since 1995 because according to the data from 2018, there are a number of territories in the territory of Lithuania where the naturalness of the landscape has increased as, the damaged territories have been restored or have recovered naturally. These trends indicated both an increase and decline in the naturalness index values used for land cover characterisation. Significant changes in land cover in the direction of renaturalisation, deforestation and urbanisation are likely to take place in these areas in the near future. The data obtained on the basis of this analysis can be further identified as problem areas in Lithuania, where changes are primarily taking place, and a review of spatial planning documents should be undertaken in order to reduce the negative naturalness of the landscape and ensure the accuracy of the green course. Our study can contribute to more effective monitoring of landscape changes across the country and the selection of appropriate measures to manage intensive land-cover change.

Currently, there is insufficient monitoring of landscape protection in Lithuania, and there is no monitoring of the implementation of the decisions of landscape-planning documents. Although the State Environmental Monitoring Programme records and analyses the progress of changes in the landscape, the data are not properly used and are mostly buried in reports. The results of the analysis of land-cover change should be used as a basis for proposals to management authorities, land-owners, and land-users. The results of landscape monitoring, specialised research and planning should form the basis for the development of landscape-information databases, specialised financial programmes, the establishment and use of funds, and the legal and institutional regulation of landscape protection and management. In order to ensure the sustainability, vitality and ability of natural and cultural landscapes to fulfil their ecological (provisioning, maintaining, regulating), social, economic and other functions, it is necessary to monitor the evolution of the relationship between natural, bio-productive and urbanised areas, and to strive to achieve an optimal one, based on scientifically valid criteria.

**Author Contributions:** Conceptualisation, A.J. and D.V.; data analysis, A.J.; writing—original draft preparation A.J. and D.V. All authors have read and agreed to the published version of the manuscript.

**Funding:** This research received no external funding. This article is part of A.J. dissertation plan.

**Data Availability Statement:** Data available on request.

**Acknowledgments:** The authors are grateful to two anonymous reviewers for their careful reading of the manuscript and valuable comments and suggestions.

**Conflicts of Interest:** The authors declare no conflict of interest.

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
