# Peer review of "Assessing Landscape Instability through Land-Cover Change Based on the Hemeroby Index (Lithuanian Example)"

_land, doi:10.3390/land11071056_

Round 1

Reviewer 1 Report

The paper very precisely documents the development of the country and classifies the changing conditions due to the ongoing intensification of the agricultural land of Lithuania. At the same time, the authors used a suitable methodological tool to evaluate these changes on the basis of CLC data. 

1.       The main topic of the work is the evaluation of trends in land changes due to various forms of intensity of use in the selected time interval.

2.       I consider the work to be a contribution in the field of science, mainly due to precise methodological preparation and a very good orientation in the issue.

3.       The work presents a model of spatial - map expression of trends in land use using the hemeroby index.

4.       It may also be appropriate to use older data to compare country changes with the present.

5.       The conclusions are formulated appropriately in accordance with the achieved results.

6.       The sources of literature used are appropriate.

7.       The tables and figures are processed well and have a good informative value.

Author Response

Author's Reply to the Review Report (Reviewer 1)

Point 1: The paper very precisely documents the development of the country and classifies the changing conditions due to the ongoing intensification of the agricultural land of Lithuania. At the same time, the authors used a suitable methodological tool to evaluate these changes on the basis of CLC data. 

Point 2: The main topic of the work is the evaluation of trends in land changes due to various forms of intensity of use in the selected time interval.

Point 3: I consider the work to be a contribution in the field of science, mainly due to precise methodological preparation and a very good orientation in the issue.

Point 4: The work presents a model of spatial - map expression of trends in land use using the hemeroby index.

Point 5: It may also be appropriate to use older data to compare country changes with the present.

Point 6: The conclusions are formulated appropriately in accordance with the achieved results.

Point 7: The sources of literature used are appropriate.

Point 8: The tables and figures are processed well and have a good informative value.

Response 1-8: We sincerely appreciate all valuable comments and and a positive review of our article. Your feedback encourages us to keep going and to continue to improve my knowledge in the field of landscape ecology. After reviewing the all reviewer’s notes, the article will be reviewed by English language professionals.

Reviewer 2 Report

This is an interesting paper discussing the instability of the plant cover/land use in Lithuania observed over a Long-Term period (23 years) of environmental changes. The analysis presents an interesting method including the concept of hemeroby to observe trends of changes, from naturalized to anthropogenic systems.

The text is almost all clear, although the authors must read it again by to check for typos, a few missed verbs, the possibility to make more concise and incisive some sentences.

I am not sure that the reference citation style is correct, because the reference list is in alphabetical order while usually should be numbered in order of citation in the text.

Line 24-26 = check the first sentence that is not written clearly

Line 66 = it seems that the article can start from this point; the previous parts are very general and should be made more concise (2 or 3 sentences?) to set the point of this article

Line 70 = I suggest deleting the word ‘Recently’ as the proposing papers are dated 1995-1997

Line 99 = add ‘.’

Line 102 = change ‘foreign’ with ‘international’

Line 116 = delete the second ‘At the beginning of 2020, ‘

Line 120 = ‘forests’ is quite vague and should be detailed (type of prevalent trees, monospecific or mixed forest, etc): are there references to the current landscape (vegetation, geomorphology, etc) ?

Lines 141-147 = are these sentences essential here? Again, these are very general statements, and it does not seem matter of ‘Study area’ within the ‘Materials and Methods’

Line 152 = when you write “anthropogenic instability of the landscape”, I wonder if the term ‘landscape’ is correct in this point. It is most properly the ‘environment’, or the ‘land cover’, the subject of your investigation, and I suggest you check all the text and change ‘landscape’ with ‘environment’ or ‘land cover’ where it is more appropriate.

Line 172 = 1.3 Km

Line 204 = Kk ?

Lines 230-232 = several labels have been described just a few lines before this paragraph, so one can avoid their repetition here

Line 249 = are there some italics?

Line 279 = please consider that you mention 4 classes of land cover/uses : 1 Agrarian change, 2. Deforestation etc with the same numbers (1 to 4) of the Landscape Instability classes just mentioned; this causes confusion. Moreover, it is not clear the relationships between the two groups of classes, and on which basis the second group have ‘ the greatest significance’…. (references?)

Table 4 = what are H1 to H7? It should be briefly reported in the caption. Why H1 is 00 everywhere?

Figure 3 = I am not sure to have read correctly, but here there is the naturalness with positive values and anthropization with negative values, isn’t it? This seems the contrary of what is shown in other parts of the paper.

Line 335 = report the ‘averaged amplitude of changes’
Line 382-383 = check XX.X% ? In general, it is not useful to repeat the same data that are shown in tables
Line 395 = the last sentence is not clear : “The processes of de-
395 forestation and urbanization in a smaller scale, but are also recorded in places.”
Lines 446-451 = Is this the aim of the paper (and must be moved to the Introduction)? Or reformulate this paragraph in the form of a ‘discussion session’.
Line 475 = this sentence should be more precise (what results? What method?)

Author Response

Author's Reply to the Review Report (Reviewer 2)

Point 1: This is an interesting paper discussing the instability of the plant cover/land use in Lithuania observed over a Long-Term period (23 years) of environmental changes. The analysis presents an interesting method including the concept of hemeroby to observe trends of changes, from naturalized to anthropogenic systems.

The text is almost all clear, although the authors must read it again by to check for typos, a few missed verbs, the possibility to make more concise and incisive some sentences.

I am not sure that the reference citation style is correct, because the reference list is in alphabetical order while usually should be numbered in order of citation in the text.

Response 1: We sincerely appreciate all valuable comments and suggestions, which helped us to improve the quality of the article. Our responses to the Reviewers’ comment are described below in a point-to-point manner.

Thank you for the very useful remark about the reference citation style. This part of the was reviewed. After reviewing the reviewer’s note, the article was be reviewed by English language professionals.

Point 2: Line 24-26 = check the first sentence that is not written clearly

Response 2: This remark has been taken into account in principle.

Point 3: Line 66 = it seems that the article can start from this point; the previous parts are very general and should be made more concise (2 or 3 sentences?) to set the point of this article

Response 3: This observation has been taken into account. The introduction has been revised, sentences have been reworded and shortened, and some have been dropped in order to make the aim and objectives of this paper as clear as possible. The introduction has not been radically restructured as suggested by the esteemed reviewer into 2-3 sentences, the authors consider it relevant to identify some previous research.

Point 4: Line 70 = I suggest deleting the word ‘Recently’ as the proposing papers are dated 1995-1997

Response 4: This remark has been taken into account and word “Recently” was deleted.

Point 5: Line 99 = add ‘.’

Response 5: This remark has been taken into account.

Point 6: Line 102 = change ‘foreign’ with ‘international’

Response 6: This remark has been taken into account.

Point 7: Line 116 = delete the second ‘At the beginning of 2020, ‘

Response 7: This remark has been taken into account.

Point 8: Line 120 = ‘forests’ is quite vague and should be detailed (type of prevalent trees, monospecific or mixed forest, etc): are there references to the current landscape (vegetation, geomorphology, etc) ?

Response 8: This observation has been taken into account. The text has been revised according to your comment to indicate which type of tree stands are dominant in Lithuania. This paper does not analyse in detail which type of forests are undergoing change, but future work will consider this at a more detailed level.

Point 9: Lines 141-147 = are these sentences essential here? Again, these are very general statements, and it does not seem matter of ‘Study area’ within the ‘Materials and Methods’

Response 9: This remark has been taken into account. Lithuania has a National Landscape Management Plan (NLMP), which is an important document for shaping the country's landscape, so it is important to briefly present the authors' views on this document. The paragraph has been corrected. Land cover changes are also analysed in different landscape management zones, showing how planning measures can influence changes.

Point 10: Line 152 = when you write “anthropogenic instability of the landscape”, I wonder if the term ‘landscape’ is correct in this point. It is most properly the ‘environment’, or the ‘land cover’, the subject of your investigation, and I suggest you check all the text and change ‘landscape’ with ‘environment’ or ‘land cover’ where it is more appropriate.

Response 10: This observation has been taken into account. The text of the article was reviewed and ‘landscape’ was changed with ‘land cover” where it was more appropriate. Some of the terms that raised questions are used in Lithuania and incorrect translation to English due to language aspects. Moreover, in Lithuania, the scientific (analytical) concept of landscape is widely used, in which landscape is understood as a combination of natural and anthropogenic components of the Earth's surface, which have a defined spatial (territorial) expression, having a variety of interconnected relationships, and an image expressing their integration. Landscape in Lithuania is used both in various scientific works and in legislation as a broader phenomenon encompassing not only land cover.

Point 10: Line 172 = 1.3 Km

Response 10: This remark has been taken into account.

Point 11: Line 204 = Kk ?

Response 11: This remark has been taken into account.

Point 12: Lines 230-232 = several labels have been described just a few lines before this paragraph, so one can avoid their repetition here

Response 12: This remark has been taken into account.

Point 13: Line 249 = are there some italics?

Response 13: This remark has been taken into account.

Point 14: Line 279 = please consider that you mention 4 classes of land cover/uses : 1 Agrarian change, 2. Deforestation etc with the same numbers (1 to 4) of the Landscape Instability classes just mentioned; this causes confusion. Moreover, it is not clear the relationships between the two groups of classes, and on which basis the second group have ‘ the greatest significance’…. (references?)

Response 14: Thanks for the very useful remark. In fact, all we wanted to do here was to highlight the most prominent land cover processes that have occurred in Lithuania. We realise that the numbering was not a good choice, so we have changed the layout to bullet points without numbering them. However, we would like to clarify that the instability classes mentioned in Table 3 are not directly related to the processes that took place in the land cover, because it is not only the process itself that is important for the identification of instability classes, but also its frequency and sequence.

Point 15: Table 4 = what are H1 to H7? It should be briefly reported in the caption. Why H1 is 00 everywhere?

Response 15: Thanks for the very useful remark. The header has been supplemented with information on the data and values in the table.

Point 16: Figure 3 = I am not sure to have read correctly, but here there is the naturalness with positive values and anthropization with negative values, isn’t it? This seems the contrary of what is shown in other parts of the paper.

Response 16: Thank you for your observation. This graph illustrates the overall change in land cover between 1995 and 2018. A negative value indicates that the total area of these sites in the country has decreased. The minus sign on this chart does not indicate anthropogenisation.

Point 17: Line 335 = report the ‘averaged amplitude of changes’

Response 17: The sentence has been revised.

Point 18: Line 382-383 = check XX.X% ? In general, it is not useful to repeat the same data that are shown in tables

Response 18: This remark has been taken into account. The text has been stripped of superfluous information that the reader can see in the graphic.

Point 19: Line 395 = the last sentence is not clear: “The processes of de- 395 forestation and urbanization in a smaller scale, but are also recorded in places.”

Response 19: This remark has been taken into account. Corrections have been made.

Point 20: Lines 446-451 = Is this the aim of the paper (and must be moved to the Introduction)? Or reformulate this paragraph in the form of a ‘discussion session’.

Response 20: This remark has been taken into account. Corrections have been made and part of the paragraph was reformulated.

Point 21: Line 475 = this sentence should be more precise (what results? What method?)

Response 21: This remark has been taken into account. Corrections have been made and part of the paragraph was reformulated. A paragraph has been added with more specific examples of the use of the results of this Article.

Reviewer 3 Report

The article “Assessing landscape instability through land cover change based on Hemeroby Index (Lithuanian example)” is based on the comparison of land use at two time points. For this, it uses the Hemeroby Index to assess the quality of its temporal evolution, thus allowing to identify socio-economic trends and potential environmental problems.

The Introduction is well developed and addresses the importance of conducting this study. Other works carried out that used the Hemeroby Index can be added to the introduction. It would be useful for the reader to present the advantages of using this index and not any other.

Figure 1. The colors must be more diversified to perceive the correct correspondence with the legend.

In topic 2.1. Study area, I suggest that the authors add important information about the territory studied, such as minimum and maximum temperatures, as well as precipitation variations.

Figure 2 lacks fundamental aspects such as the north orientation and the scale of the maps. This should be reviewed on all maps.

The remaining methodology seems to me adequate to the objectives of this study.

In Figure 6, it is not necessary to show “4 land-cover changes”, because “3 land-cover changes” also does not show results.

The letters in figure 7 must be enlarged and use the same font as the text.

The Discussion is generally well presented, however, the comparison with other studies is poor.

Line 452-457: If the authors are talking about the results, then this text should change the topic. If they are speaking in general terms, without assuming the data presented in the results, then they should be accompanied by bibliographic citations.

Conclusions begin by presenting reasoning that best fits the Discussion. No bibliographic references are required in this topic. However, in general, the authors present the main results of their investigation. As a suggestion, I ask that the importance of this study for political power be added, namely what ideas contribute to improving the quality of the landscape or avoid future problems.

Finally, authors must homogenize all citations and adopt the Land Journal's reference scheme. I also miss the recent information search. I see that the articles cited are a little old. This should be improved with the possible revision of this article.

Author Response

Author's Reply to the Review Report (Reviewer 3)

Point 1: The article “Assessing landscape instability through land cover change based on Hemeroby Index (Lithuanian example)” is based on the comparison of land use at two time points. For this, it uses the Hemeroby Index to assess the quality of its temporal evolution, thus allowing to identify socio-economic trends and potential environmental problems.

Response 1: We sincerely appreciate all valuable comments and suggestions, which helped us to improve the quality of the article. Our responses to the Reviewers’ comment are described below in a point-to-point manner. After reviewing the reviewer’s note, the article was be reviewed by English language professionals.

Point 2: The Introduction is well developed and addresses the importance of conducting this study. Other works carried out that used the Hemeroby Index can be added to the introduction. It would be useful for the reader to present the advantages of using this index and not any other.

Response 2: Thank you for your valuable comment on how to strengthen the presentation of the Hemeroby approach. In the introduction, we also mention the Hemeroby method, but we have added the advantages and uniqueness of this index.

Point 3: Figure 1. The colors must be more diversified to perceive the correct correspondence with the legend.

Response 3: This remark has been taken into account. The colours have been changed to be more similar to the meaning of the land cover.

Point 4: In topic 2.1. Study area, I suggest that the authors add important information about the territory studied, such as minimum and maximum temperatures, as well as precipitation variations.

Response 4: This remark has been taken into account. Important information on average annual rainfall, as well as temperature and natural phenomena about the study areas was added. Although this work does not deal directly with climate, looking at the current situation, we can see that climate is changing in line with landscape change.

Point 5: Figure 2 lacks fundamental aspects such as the north orientation and the scale of the maps. This should be reviewed on all maps.

Response 5: This remark has been taken into account. All maps have been revised and updated with key elements.

Point 6: The remaining methodology seems to me adequate to the objectives of this study.

Response 6: Thank you for your opinion!

Point 7: In Figure 6, it is not necessary to show “4 land-cover changes”, because “3 land-cover changes” also does not show results.

Response 7: Thank you for your comment. We have taken it into account and changed the timeline to make it clearer and to show the smallest land-cover changes as well.

Point 8: The letters in figure 7 must be enlarged and use the same font as the text.

Response 8: This remark has been taken into account. The text size and font have been aligned according to the all text parameters.

Point 9: The Discussion is generally well presented, however, the comparison with other studies is poor.

Response 9: Thank you for your valuable comment. We have added to the discussion section with references to other studies that have analysed land cover change and applied naturalness indices.

Point 10: Line 452-457: If the authors are talking about the results, then this text should change the topic. If they are speaking in general terms, without assuming the data presented in the results, then they should be accompanied by bibliographic citations.

Response 10: Thank you for your comment. We have clarified the paragraph, as it refers to the results of the study. We hope that as we write more scientific articles, our experience in writing the discussion part will increase.

Point 11: Conclusions begin by presenting reasoning that best fits the Discussion. No bibliographic references are required in this topic. However, in general, the authors present the main results of their investigation. As a suggestion, I ask that the importance of this study for political power be added, namely what ideas contribute to improving the quality of the landscape or avoid future problems.

Response 11: Thanks for the very useful remark. There is currently a lot of attention being paid to forests in Lithuania, but the landscape in general is undervalued. Based on the findings of the research, the paper also offers suggestions for legislation and policy-making institutions on how to improve the quality of the landscape or avoid future problems.

Point 12: Finally, authors must homogenize all citations and adopt the Land Journal's reference scheme. I also miss the recent information search. I see that the articles cited are a little old. This should be improved with the possible revision of this article.

Response 12: Thanks for the very useful remark. Reference list was reviewed according to the Land Journal’s reference requirements. The text has been revised and expanded to cite more recent literature and studies that add more value.

Round 2

Reviewer 3 Report

The authors did not point out the changes introduced in the text and ignored some suggestions to improve the quality of the article.

For example, where is the scale of the maps shown?

I suggest that the authors read the previous comments again and mark all the changes made in the text so that it can be easier to decide on this document.

The bibliographic references also do not conform to the journal Land standard.

Author Response

Author's Reply to the Review Report (Reviewer 3). Second round.

We sincerely thank you for your patience in evaluating the article. We have reviewed the article again and taken your comments into account. Attached is the article where the track-change shows all the changes. 

I would also like to point out that the English version of the article has been professionally checked. I can provide proof of this if required.

All maps used in this article now have north direction and scale elements. 

The references were also arranged as required. 

I very much hope that this time everything will go well and that we will be able to publish the article.